# Cell Cycle, Telomeres, and Telomerase in *Leishmania* spp.: What Do We Know So Far?

**DOI:** 10.3390/cells10113195

**Published:** 2021-11-16

**Authors:** Luiz H. C. Assis, Débora Andrade-Silva, Mark E. Shiburah, Beatriz C. D. de Oliveira, Stephany C. Paiva, Bryan E. Abuchery, Yete G. Ferri, Veronica S. Fontes, Leilane S. de Oliveira, Marcelo S. da Silva, Maria Isabel N. Cano

**Affiliations:** 1Telomeres Laboratory, Department of Chemical and Biological Sciences, Biosciences Institute, São Paulo State University (UNESP), Botucatu 18618-689, Brazil; lhc.assis@unesp.br (L.H.C.A.); debora.dede@gmail.com (D.A.-S.); me.shiburah@unesp.br (M.E.S.); bcd.oliveira@unesp.br (B.C.D.d.O.); stephany.paiva@unesp.br (S.C.P.); vs.fontes@unesp.br (V.S.F.); leilane.oliveira@unesp.br (L.S.d.O.); 2DNA Replication and Repair Laboratory (DRRL), Department of Chemical and Biological Sciences, Biosciences Institute, São Paulo State University (UNESP), Botucatu 18618-689, Brazil; bryan.abuchery@unesp.br (B.E.A.); yete.gambarini@unesp.br (Y.G.F.)

**Keywords:** *Leishmania* spp., leishmaniases, cell cycle, telomeres, telomerase

## Abstract

Leishmaniases belong to the inglorious group of neglected tropical diseases, presenting different degrees of manifestations severity. It is caused by the transmission of more than 20 species of parasites of the *Leishmania* genus. Nevertheless, the disease remains on the priority list for developing new treatments, since it affects millions in a vast geographical area, especially low-income people. Molecular biology studies are pioneers in parasitic research with the aim of discovering potential targets for drug development. Among them are the telomeres, DNA–protein structures that play an important role in the long term in cell cycle/survival. Telomeres are the physical ends of eukaryotic chromosomes. Due to their multiple interactions with different proteins that confer a likewise complex dynamic, they have emerged as objects of interest in many medical studies, including studies on leishmaniases. This review aims to gather information and elucidate what we know about the phenomena behind *Leishmania* spp. telomere maintenance and how it impacts the parasite’s cell cycle.

## 1. Introduction

Leishmaniases are among the poverty-related endemic diseases. They are well-known to cause a wide spectrum of clinical manifestations and their harsh incidences in East Africa, the Indian subcontinent, and Latin America, where approximately one million new diagnostics are expected yearly [1]. The disease is vector-induced and caused by more than twenty species of the *Leishmania* genus, protozoan parasites that belong to the Trypanosomatidae family [1]. The invertebrate host is a phlebotomine insect that is infected during a blood meal with amastigote forms. Inside the insect digestive system, amastigotes transform into procyclic promastigotes, which are noninfective but highly proliferative forms. Procyclics eventually migrate to the proboscis and differentiate into infective metacyclic promastigote forms [2,3]. The transmission to humans occurs when a preinfected insect (female phlebotomines) regurgitates during its blood meal infective but nonproliferative forms (metacyclic promastigotes) into the mammalian host skin. Afterward, metacyclics are phagocytosed by neutrophils or macrophages, and further inside the phagolysosomes, the parasites undergo a series of morphogenetic modifications, leading to the formation of amastigotes. The amastigotes then multiply and reach the bloodstream, causing the initiation of clinical manifestations. A new cycle of infection can occur when infected macrophages are ingested by other female phlebotomines (Figure 1).

Different species of *Leishmania* associated with the host biology and vector factors can lead to different clinical symptoms, ranging from light and self-cured cutaneous lesions (e.g., *Leishmania major*) to life-threatening visceral complications (e.g., *Leishmania infantum*) [2]. The disease is endemic in more than 60 countries, with East Africa and the Indian subcontinent remarkably impacted. Although a significant decrease in the number of cases has been reported in the past years, mainly due to efforts in vector control, leishmaniasis is still on the top priority list for developing new treatments. Such a scenario is typified, because toxic antiparasitic drugs (e.g., antimonial pentavalent and amphotericin B) are still in use nowadays. Additionally, other therapeutic approaches such as vaccination have been shown to be not adequate in eradicating the disease [4]. Thus, there is urgency for new specific and efficient treatments against leishmaniasis. For this purpose, several studies on *Leishmania* spp. have been conducted to elucidate the different aspects of parasite biology, including genetic studies and full sequencing of the parasite genome [4,5,6]. Among these studies, trypanosomatid telomere biology has awakened great interest in the scientific community, as telomeres are essential for genome stability and cell cycle progression. Furthermore, some components involved in parasite telomere maintenance, such as the telomerase RNA component and members of the shelterin-like telomeric complex (e.g., TRF, RPA-1, and Rbp38), are unique and present parasite-specific features relative to their hosts [7,8,9,10,11,12,13,14]. Therefore, new advances in understanding and exploring the parasite telomeric environment may reveal how these and other unknown parasite factors could be used as potential and specific targets for drug development [8,9,10,11,12,13,14,15,16,17,18,19].

Telomeres are the physical ends of eukaryotic linear chromosomes. Structurally, they are an association between tandemly repeated noncoding DNA sequences and nucleoproteins forming complexes at the end of chromosomes [20]. Telomeric DNA is composed of a double-stranded sequence (one of them rich in cytosine, the other in guanine) and a G-rich single strand that protrudes toward the ends of the chromosome, known as the 3′ overhang [20]. In humans and other vertebrates, the repeated telomeric sequence is 5′-TTAGGG-3′, approximately 3–15 kb in length and associated with a six-telomeric protein complex called shelterin [21,22,23,24,25]. These arrangements of DNA and proteins influence cell homeostasis. They are crucial to cell cycle maintenance, being decisive in important cellular processes such as cell aging, genome integrity, and maintenance of the nuclear arrangement [26,27,28]. It is known that, lengthwise, telomeres tend to shorten after each cell division due to the inability of DNA polymerase to complete replications in the lagging strand of linear chromosomes [29], culminating in the progressive loss of telomere repeats. Thus, telomeres act as a molecular clock. If its size reaches critical levels, it can lead to early/unprogrammed cell senescence or even activate local DNA damage repair, eventually triggering a mitotic catastrophe [14,30,31,32]. In most organisms, including *Leishmania* spp., telomeric DNA is elongated by telomerase. This specialized reverse transcriptase forms a ribonucleoprotein (RPN) complex [31], whose function is strictly regulated throughout the cell cycle [17,21,32]. In that sense, telomerase, telomeres, and their associated proteins are now recognized as potential drug targets [33]. In the face of the critical medical relevance of leishmaniases, peculiarities associated with the cell cycle, telomeres dynamics, and telomerase have reached the mainstream studies on parasite biology. This review aims to shed light on what we know about the phenomena behind telomere maintenance and how it impacts parasites’ cell cycle and survival.

## 2. *Leishmania* spp. Cell Cycle

The cell cycle comprises a repeating series of events encompassing growth, DNA synthesis, and cell division. For most eukaryotes (including *Leishmania* spp.), the cell cycle follows a single pattern of organization consisting of the following phases: G1 (gap 1), S (DNA synthesis), G2 (gap 2), M (mitosis), and C (cytokinesis). In addition, the G0 state is sometimes included as part of the cell cycle. However, it is noteworthy that cells in the G0 state are not stimulated to proliferate [34]. In other words, cells in the G0 state are not yet committed to genome replication and cell division. Interestingly, the infective forms of most parasitic microorganisms are at the G0 state (e.g., metacyclic promastigotes of *Leishmania* spp.) [35,36] (Figure 2), which makes us wonder if, in these organisms, proliferation and infection are mutually exclusive events.

In *Leishmania* spp., as in most trypanosomatids, the G1 phase corresponds to the larger proportion of the cell cycle, while the other phases vary slightly in duration [42,43]. No studies have evidenced the metabolic changes and main events during the G1 phase in *Leishmania* spp. However, compared with the other trypanosomatids, we can infer that, in the G1 phase, there is an increase in the transcription rate and intense protein synthesis of the factors related to DNA replication [44,45]. Moreover, in the G1 phase occurs the establishment of a divergent prereplication protein complex at specific sites on the chromosomes called replication origins, which can give rise to a replication bubble [45].

The firing of replication origins starts the S phase, which briefly consists of the reliable replication of DNA molecules. Usually, eukaryotes have many replication origins per chromosome. However, in *Leishmania* spp., the number of replication origins per chromosome is an issue that generates debate. Six years ago, Marques et al., (2015) [46] used a marker frequency analysis coupled with deep sequencing (MFA-seq) to evidence that *L. major* replicates each of its chromosomes during the S phase using a single replication origin. However, in a more recent study, da Silva et al., (2020) [47] used mathematical equations to reveal that it is improbable that *L. major* uses only a single origin per chromosome during the S phase. The explanation for these discrepancies is that the MFA-seq approach is not sensitive enough to identify all replication origins [47]. Nevertheless, further sensitive assays are needed to establish how many origins are used per chromosome during the S phase in *Leishmania* spp.

In model eukaryotes, the G2 phase is characterized by the duplication of centrioles and other cytoplasmic organelles [48]. In trypanosomatids, homologs of the proteins described in model eukaryotes involved with centriole biogenesis are associated with the basal body and flagellum biogenesis [49,50]. Furthermore, based on studies with other organisms [51], we can infer that, in the G2 phase, *Leishmania* spp. increase the rate of transcription and resumption of intense protein synthesis, which are necessary for the completion of cell division. This entire process results in an increase in cell volume and size [37,40].

During the M phase, *Leishmania* spp. and all other trypanosomatids do not disassemble their nuclear envelope and perform a closed mitosis process, which is organized by spindle pole body-like structures [52]. Moreover, due to the absence of the N-terminal portion and globular domain of histone H1 and the absence of phosphorylation on serine 10 of histone H3 (H3S10), *Leishmania* spp. and other trypanosomatids are unable to condensate their chromosomes into 30-nm fibers [53,54].

In mammals, a curious feature worth being highlighted is that mitosis and cytokinesis overlap, since cytokinesis begins before the mitotic chromosome segregation is complete. However, *Leishmania* spp. seem not to strictly follow this premise. Once mitosis ends, *Leishmania* spp. undergo a rapid remodeling in shape, first growing in length and then in width prior to cytokinesis, which ends cell division [52,55]. Although it is challenging, all these peculiarities related to the cell cycle phases may provide new routes toward the search for suitable targets for parasite cell cycle interventions aiming at their elimination.

A set of events that deserve being highlighted during the *Leishmania* spp. cell cycles refers to the replication and segregation of the kinetoplast. The coordination of these events throughout the *Leishmania* cell cycle does not follow those equivalents in model eukaryotes, where mitochondrial DNA replicates at any cell cycle stage [56,57]. The nuclear and kinetoplast S phase occurs almost simultaneously, but the effective segregation of these organelles can occur at different periods according to the species analyzed. Many studies have established a pattern of segregation for the kinetoplast relative to the nucleus in some species of *Leishmania* [37,38,39,40,41].

One of these studies characterized the main morphological events of the cell cycle of *L. mexicana* promastigotes [37]. Wheeler et al., (2011) [37] described the cell cycle phase durations and established a duplication order for the kinetoplast and nucleus. Two years later, da Silva et al., (2013) [40] characterized the length of the *L. amazonensis* cell cycle phases and evidenced that the promastigote forms present two distinct modes of kinetoplast segregation relative to the nucleus, following a distinct temporal order in different proportions of the cells. Using DAPI staining and EdU (5-ethynyl-2′-deoxyuridine) to monitor, respectively, the segregation of DNA-containing organelles and DNA replication, the authors found a curious feature: 65% of the dividing promastigotes duplicate the kinetoplast before the nucleus, and the remaining 35% do the opposite or duplicate both organelles concomitantly. This finding corroborates another study carried out with *L. donovani* promastigotes, where the authors found that about 80% of the cells divide the nucleus before kinetoplast, and the remaining 20% do the opposite [41]. In other words, *L. donovani* and *L. amazonensis* exhibit a nonfixed pattern of nucleus and kinetoplast segregation. In fact, when we compare the cell cycle among different *Leishmania* spp., the order and timing of the kinetoplast and nucleus division are not consensual and cannot be generalized [37,38,39,40,41]. 

For instance, *L. mexicana*, *L. major*, and *L. tarentolae* exhibit fixed patterns of kinetoplast and nucleus segregation. However, *L. mexicana* segregates its kinetoplast predominantly after the nucleus [37], while *L. major* and *L. tarentolae* do the opposite [39,41] (Figure 2). A possible explanation for these different behaviors resides in the fact that, although belonging to the same genus, these parasites show considerable phylogenetic distance [58]. In other words, this phylogenetic divergence may reflect possible species-specific differences relative to kinetoplast segregation, suggesting that some *Leishmania* spp. have less stringent control over the order of division of their DNA-containing organelles (nucleus and kinetoplast). More studies are needed to uncover the potential players involved in controlling cell division and organelle segregation, since some of them could be explored for precise interventions related to the parasite cell cycle.

## 3. *Leishmania* spp. Telomeres

In *Leishmania* spp., telomeres constitute short noncoding repetitive sequences of 5′ TTAGGG 3′ [16,59], except for *Leishmania braziliensis,* which, in addition to the conventional sequence, is also observed the presence of 5′-CCTAACCCGTGGA-3′ sequences at the ends of some chromosomes [60]. While, in *L. amazonensis*, the 3′ G overhang has an approximate size of 12 nt long (5′-GTTAGGGTTAGG-3′), in *L. donovani* and *L. major*, the 3′G overhang is a 9-nt sequence composed of 5′ GGTTAGGGT 3′ [61]. The studies performed by Genest & Borst [62] described that the length of the *L. tarentolae* and *L. major* telomeres increase over time by approximately 1 bp per population doubling. Similar results were obtained by Oliveira et al., (2021) [15], where telomeres of the *L. amazonensis* procyclic promastigotes increased during continuous passages. It was also observed that amastigote telomeres (Figure 1, numbers 1, 6, and 7) are shorter compared to procyclic (Figure 1, number 3) and metacyclic promastigotes (Figure 1, numbers 4 and 5).

Near telomeres, there are the subtelomeric sequences, which, in *Leishmania*, are composed of 100-bp-long conserved telomere-associated sequences (LCTAS) that contain two conserved domains, the conserved sequence block 1 (CSB1) and conserved sequence block 2 (CSB2) [60]. These sequences have a high degree of conservation among the different *Leishmania* species, which may indicate their importance in chromosome segregation or as binding sites for telomeric proteins [60]. However, despite the CSB sequences and depending on the species, LCTAS also contains some nonconserved sequences and a high degree of polymorphism [59,60,63,64]. Furthermore, telomeres are commonly associated with proteins that interact directly with both telomeric strands and other telomeric proteins that can influence the telomere size by regulating the telomerase access, providing chromosome stability and protecting telomeres from degradation [65]. So far, within the *Leishmania* genus, the *L. amazonensis* telomeric complex (Figure 3) is the best-characterized.

Lira et al., (2007) [67] described a 45-kDa telomere-binding protein named LaTBP1 as a *Leishmania* double-stranded DNA-binding protein that interacts with GT-rich and telomeric DNA sequences. LaTBP1 has at least a central Myb-like DNA-binding domain containing a conserved hydrophobic cavity involved in DNA binding, a feature conserved among the proteins that bind the telomeric double-stranded DNA [67]. However, it is unclear if LaTBP1 binds DNA as a monomer like RAP1 (Repressor Activator Protein 1) in mammals or as a dimer like the TRFs. It is worth highlighting that the LaTBP1 Myb-like DNA-binding domain is related closer to the two centrally located RAP1 Myb domains than the single C-terminal TRF (Telomere Repeat-binding Factor) Myb domain [67]. 

Later, a TRF homolog was described in *L. amazonensis* (LaTRF) by da Silva et al., 2010 [68]. This protein presents 82.5 kDa, shares structural features with mammalian TRF1 and TRF2, and is highly similar to *Trypanosoma brucei* TRF. It also contains a Myb-like DNA-binding domain that allows it to bind double-stranded telomeric DNA. Apparently, LaTRF is little abundant, and it is localized in the nucleus [68].

Some proteins that bind single-stranded telomeric DNA were identified in *L. amazonensis* [69]. RNA-binding protein 38 (LaRbp38) is one of those. It is a 38-kDa protein with a noncanonical binding site with an affinity to both single- and double-stranded G-rich telomeric DNA and GT-rich kinetoplast DNA (kDNA) [11,70]. LaRbp38 has dual cellular localization, and in synchronized *Leishmania* promastigotes, it shuttles from the mitochondria to the nucleus at the late S and G2 phases via importin alpha [11]. LaRbp38 was first identified in a telomerase-positive extract together with RPA-1, LCalA, and DNA polymerase alpha [69]. The LaRbp38 ability to bind telomeres and kDNA, and the fact that this protein is involved with kDNA stability and replication, suggest that it may also be involved with telomere elongation. However, further assays are necessary to check the veracity of this hypothesis.

Replication protein A1 (LaRPA-1) was another protein pulled down with LaRbp38 [71]. LaRPA-1 is a nuclear protein of 51 kDa that presents a single N-terminal OB-fold domain. This protein binds the telomeric G-rich ssDNA, protecting it from 3′-5′ exonucleolytic degradation. Thus, LaRPA1 is considered a potential trypanosomatid telomere end-binding protein (TEBP) that shares structural and functional features with other TEBPs described in model eukaryotes. It binds at least one telomeric repeat using an OB-fold domain, protects telomeres from exonuclease degradation, and unfolds the telomeric G-quadruplex [12,71,72]. The G-quadruplex structure is known to impair telomerase activity in humans [73,74]. Although not tested against *Leishmania* spp., drugs that specifically bind these structures could be a new source for antiparasitic therapies.

LCalA (MW 16 kDa) was the third protein identified in the parasite telomerase-positive extracts [75]. It is the first reported protozoa calmodulin-like nuclear protein that binds in vivo to the G-rich telomeric strand and the 3′ G overhang. The binding of LCalA to telomeres is calcium-dependent. Biophysical assays showed structural changes in LCalA in the presence of calcium ions, increasing the affinity of this protein to telomeres. Additionally, LCalA partially colocalizes with telomeres throughout the parasite’s cell cycle. LCalA resembles human KIP1 protein, since both share with calmodulins two EF hand domains and the affinity for calcium. Furthermore, KIP1 was shown to be involved with telomere homeostasis by its interactions with the telomerase TERT component and with shelterin member TRF2 through the EF hand domains. Whether LCalA is a functional homolog of KIP1 is a question that remains open.

Long noncoding RNAs called TERRA (telomeric repeat-containing RNA) were also described at *Leishmania* telomeres. TERRA is transcribed from the C-rich subtelomeric strand towards the end of chromosomes. Its main function is to regulate the telomere length [76]. In *L. major*, TERRA was shown to be polyadenylated and processed by trans-splicing [77]. It was also observed that the number of TERRA transcripts was higher in the infective forms of the parasite (metacyclic promastigotes) relative to the procyclic promastigote and amastigote forms [77]. Moreover, Morea et al., (2021) [77] showed that TERRA could form R-loops, suggesting that *L. major* TERRA is engaged with various cellular processes, including telomere maintenance and the regulation of telomere lengths. 

Another interesting finding related to *Leishmania* telomeres is the presence of base J (β-d-glucosyl-hydroxymethyluracil). Base J is a modified thymine that can interfere with RNA polymerase II function, and its presence abrogates DNA cleavage by restriction endonucleases [78]. This hypermodified DNA base was first described in the bloodstream form of *T. brucei* [79,80]. Later, base J was characterized in many other trypanosomatids, and in most of them, except the *Leishmania* spp., it was found spread out in different chromosome regions [81,82]. In *Leishmania* spp., ~98% of base J is found in telomeres [62], suggesting that it can be related to telomere function [83]. Furthermore, the recent results of Morea et al. (2021) [77] suggest that differences in the telomeric base J levels may control TERRA transcription in the *L. major* developmental forms and during continuous in vitro passages. 

## 4. *Leishmania* spp. Telomerase

### 4.1. Structure and Function

Telomerase is the ribonucleoprotein responsible for telomere elongation. The enzyme adds 5′-TTAGGG-3′ repetitions at the single-stranded 3′G overhangs of telomeres [83]. Thus, telomerase is thought to be essential to the regulation of telomere lengths and maintenance of genomic stability. Telomerase presents two main components, protein Telomerase Reverse Transcriptase (TERT) and Telomerase RNA (TER), minimally required for enzyme activity in vitro [84,85]. TERT is the catalytic component, and TER contains the template used by TERT to synthesize telomeric repeats. These two components form a complex with accessory proteins and are necessary for in vivo biogenesis, enzyme activity, and nucleotide addition processivity [13,32,86]. *Leishmania* spp. telomerase activity was first reported by Cano et al., (1999) [87], and enzyme purification and biochemical characterization were further described by Giardini et al., 2011 [88]. Parasite enzyme activity was detected in all three parasite life forms presenting the canonical properties of other telomerases. However, the catalysis was shown to be temperature and life-stage dependent [15]. The genes encoding the *Leishmania* spp. TERT was characterized by Giardini et al., (2006) [89], and the TER component was described later by Vasconcelos et al., 2014 [7]. 

In trypanosomatids, as in most eukaryotes, the TERT component is structurally composed of four domains [13]: The Telomerase Essential N-terminal (TEN), Telomerase RNA-Binding Domain (TRBD), Reverse Transcriptase domain (RT), and the C-terminal extension (CTE). The TEN domain is involved in telomerase recruitment to the telomeres and enzyme processivity [90]. The TRBD interacts with TER and is connected to the TEN domain by an unstructured linker and creates the RNA-binding pocket that binds single stranded and paired RNA [91]. Both domains can also interact with proteins that stabilize the complex and help to recruit telomerase to telomeres and regulate enzyme activity [92,93,94,95,96,97]. The RT is the catalytic core of the enzyme, which interacts with TER through the pseudoknot region [98], and it is involved in the interactions with hybrid RNA–DNA. The CTE domain stabilizes the RNA–DNA duplex, and differently from the other three domains, it is less conserved among different species [13]. *Leishmania* TERT preserves all the canonical domains found in other TERTs but shows some amino acid substitutions that are specific to the genus [89]. The knockout of *Leishmania* TERT seems to be very harmful to the parasite, because it induces a gradual decrease in cell density in the culture, apparent during G1/G0 cell cycle arrest, morphological alterations, and telomere shortening (unpublished data).

The RNA component TER varies in length and sequence, presenting a conserved secondary structure in most eukaryotes. Variations of the TER size and sequence are observed among different organisms, and they are more prominent than TERT [99], which is conserved even among different taxa. In *Leishmania* spp., TER (LeishTER) is about ~2100 nucleotides long, and the mature molecule modified by trans-splicing contains a 5′ cap, a spliced leader sequence (SL), and a 3′ polyA tail. It also presents a C/D box snoRNA domain found in other TER [7]. LeishTER is expressed at similar levels in its procyclic and metacyclic promastigote forms. The mature molecule coimmunoprecipitates and colocalizes with the TERT component in a cell cycle-dependent manner. Its secondary structure prediction shows the template sequence (5′→3′) in a single-stranded form localized near the 5′ end of the RNA molecule. LeishTER also presents a conserved TBE (Template Boundary Element) motif C[U/C]GUCA in helix II, which is responsible for interacting with the TERT TRB domain [7]. Its double knockout led to partial cell cycle arrest and increased apoptosis in procyclic promastigotes. TER KO also triggers a progressive telomere shortening during continuous parasite passages (unpublished data). A similar effect was observed in *T. brucei* TER knockouts [100].

### 4.2. Biogenesis and Mechanisms

The biogenesis of the telomerase ribonucleoprotein complex is an intricate process involving the assembly of proteins and TER with the subcellular localization of the ribonucleoprotein. Besides TERT and accessory proteins, other proteins interact transiently with telomerase. These other proteins are important players in maturation, stability, and subcellular localization [101,102].

The telomerase biogenesis starts with the transcription of TERT and TER mRNAs by RNA polymerase II. While TERT mRNA is addressed to the cytoplasm to be translated, the TER remains in the nucleus and is assembled with the accessory proteins at the 3′end—specifically, at the C/D box motif [7]. Unfortunately, no 3′end-binding protein has yet been identified in *Leishmania* spp. Still, considering the other trypanosomatids, it is known that the C/D box motif of *T. brucei* TER interacts with SNU13, a protein known to interact with the C/D box in other eukaryotes. Together with SNU13, in humans, three other proteins recognize and bind to this RNA motif: fibrillarin, NOP56, and NOP58. However, the evidence for homologs of these proteins is still missing in these parasites. In the case of TERT mRNA, in most organisms, after translation, the protein migrates from the cytoplasm to the nucleus, where the telomerase is assembled. In humans, the main proteins involved in TERT traffic are HSP90, its cochaperone p23, and two AAA^+^ ATPases (Pontin and Reptin) [101,102,103]. However, a complete understanding of the TERT traffic in *Leishmania* spp. remains to be determined. In these parasites, the HSP90 ortholog is the protein HSP83. Recently, Oliveira et al., (2021) [15] revealed an interaction with the TERT component and the importance of HSP83 for telomerase activity and telomere maintenance in *L. amazonensis*. These findings suggest that *Leishmania* spp. TERT traffic to the nucleus may resemble the human telomerase process. Furthermore, in this study, the HSP90 inhibitor 17AAG affected both the *L. amazonensis* telomere length and telomerase activity using a thousand-times less of the drug needed to inhibit HSP90 in host cells [104].

Considering the assemblance of human telomerase as an example, the location of telomerase in Cajal bodies is the key to ribonucleoprotein trafficking and recruitment to the telomeres [101,102]. This subcellular localization is derived by the CAB box sequence at the 3′end of human TER and its interaction with the TCAB1 protein. Otherwise, in the context of *Leishmania*, nothing is known about the existence of Cajal bodies [105]. The closest information is related to the identification of the MTAP protein, a TCAB1 protein ortholog of *T. brucei* [106] that can interact with TER [107]. Therefore, the biogenesis of telomerase in *Leishmania* spp. still has many gaps to be replenished.

Once the telomerase RNP complex is assembled, it is recruited preferentially to the shortest telomeres. The ability to add more than one repeat at one telomere is called repeat addition processivity (RAP) [108]. In humans, telomerase activity is regulated by shelterin subcomplex POT1-TPP1, with TPP1 being the key player [109,110,111]. These two proteins are important to decrease the dissociation rate of telomerase and help the translocation step. In *Leishmania* spp., RPA-1 seems to play the same role as POT1 at telomeres [12]. However, no ortholog of TPP1 was described in the parasite. The identification of other parasite telomerase cofactors and the characterization of this multistep process are under investigation.

Interestingly, in mammals, it is known that telomerase is possibly involved in other roles not related to telomere maintenance, such as cellular proliferation, mitochondrial activity, and gene expression regulation [112]. In *Leishmania* spp., for example, Campelo et al., (2015) [113] studied the TERT component of *Leishmania major* and observed that, besides the nucleus, it could also be found in its unique mitochondrion. In addition, parasites exposed to hydrogen peroxide showed increased TERT levels, especially in the mitochondria compartment. However, it is important to remember that parasite mitochondrion contains circular DNA (kDNA), which does not have telomeres. Therefore, it is important to demonstrate if parasite telomerase presents extra telomeric functions, since it could increase the druggable potential of this important ribonucleoprotein. 

Our belief in telomerase as a good target for antiparasitic drug development is also related to its essential role in cell homeostasis and genome stability. We must also consider the structural differences between the telomerase complex (TERT and TER) of mammals and *Leishmania* spp. [7,89] and the fact that telomerase is inactive in most mammalian somatic cells. Thus, it is reasonable to hypothesize that impairing the infective parasite forms to elongate telomeres would directly affect parasites without harming host cells.

### 4.3. Phylogenetic Context of Leishmania spp. Telomerases among Other Pathogenic Trypanosomatids

Our very little knowledge about telomerases in *Leishmania* spp. reflects the challenges involved in the genome manipulation of these parasites [114,115]. However, the recent advent of genome editing using CRISPR-Cas9 and its variations should help resolve some of these functional questions that are still lingering, as has been done for other genes and in other species [116,117,118]. In addition, the available evidence on the subject matter shows some differences in the TERT between members of the trypanosomatid family and may be due to the enzyme′s usefulness to the parasite [89,119].

Considering the importance of a high proliferative rate for the cell cycle of these parasites, the question of whether the parasite depends on an enzyme for longevity, as has been reported in other cells, is still unanswered. The knockdown of *T. brucei* telomerase induced telomere shortening without effects on the parasite growth [119]. However, the existent differences between trypanosomatids cannot be confidently extrapolated. In *Leishmania* spp., for example, the absence of the TERT and the TER minimal telomerase components seems to be very harmful to the cells (see Section 4.1).

Between the members of the *Leishmania* genus, Giardini et al., (2006) [89] found that the TERT component of *L. donovani*, *L. amazonensis*, *L. major*, and *L. braziliensis* has a nucleotide identity exceeding 90%, while sharing a little similarity with *T. cruzi* and *T. brucei*. Concerning amino acids, their sequence analysis shows over 70% identity between the *Leishmania* spp. and under 40% similarity with *T. cruzi* and *T. brucei* [89]. The *Leishmania* TER component also does not share nucleotide sequence similarities with other trypanosome TERs. They only share similarities in their secondary structures [7].

In other eukaryotes, the TERT component is also involved with noncanonical and extratelomeric functions, such as cell proliferation and apoptosis [120,121,122,123,124]. However, these functions may more likely result from evolutional adaptation events, which sometimes result in the loss and gain of certain protein domains with organism-specific functions. Hopefully, with the new approaches based on the CRISPR/Cas system, the many gaps of the *Leishmania* spp. telomerase will begin to be covered, and more details will probably be available to elucidate the telomere biology of these parasites.

## 5. Conclusions Remarks

The telomeres dynamics throughout the cell cycle are an essential phenomenon for all eukaryotic cells. Additionally, for most eukaryotes, the holoenzyme telomerase is the key behind these dynamics. When we expand this fact for single-cell parasites, we assume that understanding the differences and similarities between the pathogen and the host is an essential pathway for specific and successful treatment development. Here, we covered the knowledge available so far on *Leishmania* spp. cell cycle and telomere homeostasis, unveiling the remaining gaps and the advances reached in the last years. Among the impressive progress on the biology of these parasites are the facts that they remain in the G0 state during the infective stages and the remarkable divergence of their telomeric shelterin-like complex relative to mammals. However, these aspects are only a few examples of how this subject can be future linked to leishmaniases treatment and how far scientists are from a deeper knowledge of these peculiar eukaryote parasites.

## Figures and Tables

**Figure 1 cells-10-03195-f001:**
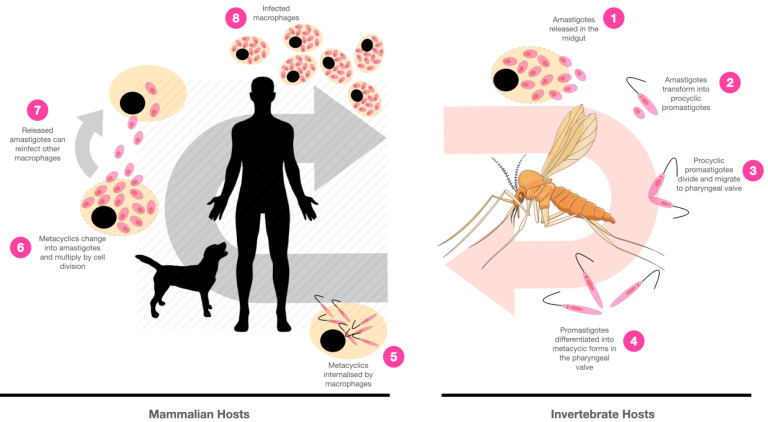
Schematic illustration of the *Leishmania* spp. infective cycle showing different parasite stages of development. Phlebotomine female sandflies get infected with amastigote forms during bloodmeals (1). Amastigotes transform into procyclic promastigotes, which will proliferate inside the invertebrate midgut (2 and 3). Promastigotes then migrate to the sandflies’ pharyngeal valve (4) and differentiate into metacyclic forms. The metacyclic forms are transferred to the mammalian hosts’ bloodstream during a new blood meal and infect macrophages and other cells from the mononuclear phagocyte system (5). Inside the macrophages, there is a metacyclic change into amastigote forms, which multiply, lyse the macrophage, and reinfect new macrophages (6 and 7). In a new cycle of infection, the infected macrophages are ingested by new phlebotomines (8). The silhouettes of man and dog and the sandfly clipart are free for use and were withdrawn from the websites HiClipart (https://www.hiclipart.com, accessed on 8 November 2021) and Gratispng (https://www.gratispng.com, accessed on 8 November 2021), respectively.

**Figure 2 cells-10-03195-f002:**
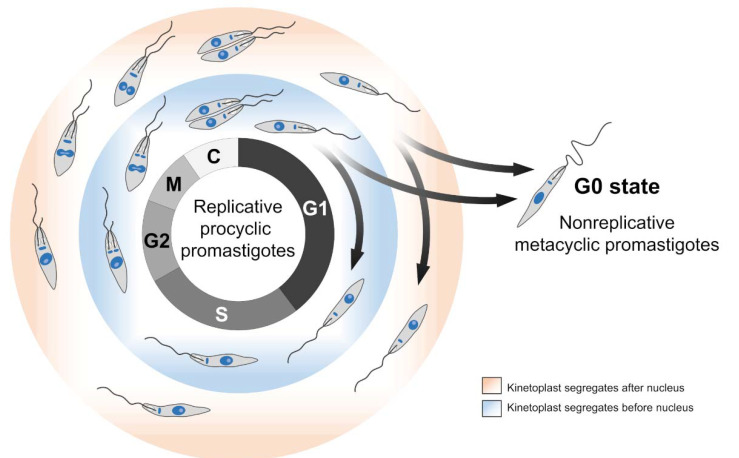
Scheme showing the two distinct patterns of kinetoplast segregation relative to the nucleus in promastigotes of *Leishmania* spp. throughout the cell cycle. Most *Leishmania* species presents one of the two kinetoplast segregation patterns presented: kinetoplast segregates after the nucleus (light red) or kinetoplast segregates before the nucleus (light blue). For instance, *L. mexicana* segregates its kinetoplast predominantly after the nucleus [37], while *L. major* and *L. tarentolae* do the opposite [38,39]. However, *L. donovani* and *L. amazonensis* exhibit these two patterns distributed in the population [40,41].

**Figure 3 cells-10-03195-f003:**
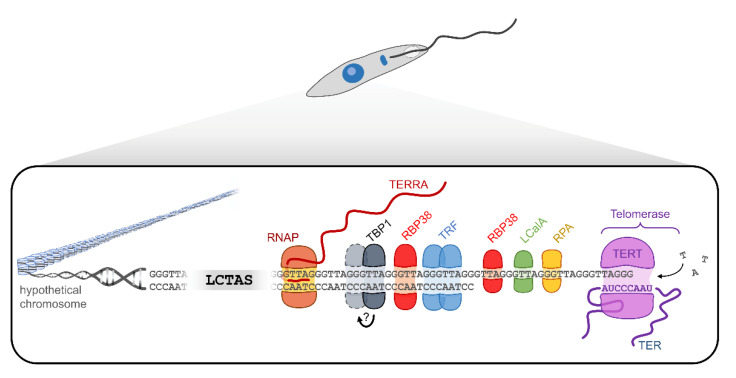
*Leishmania* spp. telomeres are formed by subtelomeric regions (LCTAS); a double-stranded region; a single-stranded protrusion (3′-G overhang); and associated proteins, such as TBP-1, TRF, Rbp38, RPA-1, and LCalA. Of note, it is not clear if TBP1 binds DNA as a monomer or dimer. The telomeres are elongated by telomerase, a ribonucleoprotein composed minimally by TERT and TER. From the C-rich strand of the telomeric region, ncRNAs called TERRAs are transcribed and appear to be involved in telomere length regulation. Adapted from da Silva et al., 2012 [66].

## Data Availability

Not applicable since this work is a review of published data.

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
