# Peer review of "Cell Cycle, Telomeres, and Telomerase in Leishmania spp.: What Do We Know So Far?"

_cells, 2021, doi:10.3390/cells10113195_

Round 1

Reviewer 1 Report

The MS is well prepared and carefully written in good English. The number of minor remarks below:

  1. line 193 – ref to Fig.1 will be useful. It’ll help unqualified readers. Especially if authors will tell kind of “amastigote's (1, Fig.1) telomeres are shorter compared to the procyclic (2, Fig.1)   and metacyclic promastigotes (3, Fig.1). I’m not sure with exact numbers, but useful to mention the numbers on Fig in the text.
  2. line 271, 276 Leishmania spp in Italic
  3. 3. line 295. (Shiburah and Cano, personal communication) May be it is “unpublished observation” if both are the co-authors of this very MS?

Line 310 (Oliveira and Cano, personal communication)    the same

  1. line 366(see topic 4.1) May be (see section 4.1)
  2. It remains not clear – what kind of treatment against parasites through telomere-protein complex could be possible. I highly recommend to delete this statement from Introduction: line 56-57 “new advances in understanding the parasite telomeric environment increase the chances of finding new and exclusive parasites' drug targets [7,8,9,10].” The other way is to add your suggestion about parasites' drug targets at the end, i.e. to underline the species to which telomerase block could be successful.
  3. lines 194-203. From this paragraph reader could have the wrong impression that Leishmania subTel areas all conservative. The LCTAS  vary in number of copies, in chromosomes positions in L. braziliensis, L. major, L.donovani, L.amazonensis.  Your own work shows the subTel variability (Genomic organization of telomeric and subtelomeric sequences of Leishmania (Leishmania) amazonensis F.F. Conte, M.I.N. Cano). There are other subTel sequences,  L. major MTAS274 for instance.  Better to clarify this paragraph.  
  4. Figure 3 and line 211 and so on. LaTBP1 name in different in the MS and figure, so it became confusing for the reader. Two myb domains are necessary for binding and how LaTBP1 is bound to Tel DNA is not clear.

  5. line 223 and so on. Different spelling for the same protein is confusing. Optional. Rbp38 and LCalA look like  assistive Tel-binding proteins for their association with Tel is temporary.  The shuttling of  Rbp38 is already mentioned in relation to inconsistent division of genome DNA and kinetoplast (lines 157-182). It would be interesting to know data about subcellular localization of Rbp38. I dare to suspect that Rbp38 involved in Tel replication. There is nothing about such a possibility in the current MS. Is there such a literature?

       The data about LCalA also incomplete. Is there the dependence of LCalA presence on Tel from the cell cycle? What is the sense of LCalA presence on Tel?

Reviewer 2 Report

Dear Editor,

The review manuscript “Cell cycle, telomeres, and telomerase in Leishmania spp.: what do we know so far?” by Luiz Henrique de Castro Assis, Débora Andrade-Silva, Mark Ewusi Shiburah, Beatriz Cristina Dias de Oliveira, Stephany Cacete Paiva, Bryan Etindi Abuchery, Yete Gambarini Ferri, Veronica Silva Fontes, Leilane Silva de Oliveira, Marcelo Santos da Silva, and Maria Isabel Nogueira Cano represents the current knowledge on telomerase and cell cycle in Leishmania species. The manuscript is very valuable and can be noticed by the audience of your journal. The authors have studied the field for several years and discovered many of the important aspects. Most of the critically relevant papers have been cited properly.  In some cases, it has deviated from the holistic process of writing a review article.  Some of the points that may improve the manuscript are suggested:

  1. Lines 389-391 “Among the interesting progress on the biology of these parasites, are the fact that they remain in the G0 state during the infective stages and the remarkable divergence of their telomeric shelterin-like complex relative to mammals.”; while G0 state of infective form of the parasites may inferred as no telomerase activity one expects to hear clearly whether telomere/telomerase can be a good molecular target for drug design against Leishmania spp. According to extra-nuclear activity of telomerase, that would also be worthful to note that it is still a significantly interesting and valuable candidate in drug design.
  2. "Figure 2. Scheme showing the two distinct patterns of kinetoplast segregation relative to the nucleus in promastigotes of Leishmania spp. throughout the cell cycle." The authors should discuss how important is this issue? Does it affect the pathogenesis of these parasites?
  3. Telomerase can also locate in mitochondria and Kinetoplasts; could this be related to pathogenicity of Leishmania species? The authors are strongly encouraged to discuss alternative functions of telomerase and its relevance.

Best regards,

Reviewer 3 Report

The article by Assis et al. is a review entitled “Cell cycle, telomeres, and telomerase in Leishmaniaspp.: what do we know so far?”, to be published in a special issue of Cells. This review aims at documenting the present knowledge of the scientific community on the various life cell cycles as well as telomere biology of the trypanosomatids in order to potentially use telomeric targets to cure Leishmaniases. In particular, the aim of these studies is to understand how telomere maintenance can impact parasite's homeostasis. Leishmaniasis has been reported as one of the most dangerous neglected tropical diseases, second only to malaria in parasitic causes of death. Possibly up to 350 million people are susceptible to be infected worldwide in a total of 98 countries.

Major comments:

* This review gives a pretty good and recent picture of the current knowledge on the various life stages as well as telomerase and telomeric proteins of several trypanosome organisms, known to be responsible for leishmaniasis, provoked by more than 20 species of Leishmania. Interestingly, the authors highlight the differences in events of the cell cycle between some of these different species of Leishmania orTrypanosoma. These differences are likely to be important when dealing with finding targets to kill the trypanosomes at a particular point of their cell cycles.

* As expected, there are much less telomeric proteins identified in trypanosomes than in the model organisms such as humans, yeast or Drosophila. The authors have written a pretty good summary of the situation concerning these telomeric proteins as well as telomerase.

* At the very end of the Abstract, the authors state “This review aims to gather information and elucidate what we know about the phenomena behind the parasite cell cycle and how telomeres maintenance impacts the parasite's homeostasis”. There are indeed observations of differences in telomere length, for instance between different life stages of the parasite, as noted by the authors, but, on the other hand, there is no real clue as to whether acting on telomerase at either one of these stages will result in parasite killing, for instance. Moreover, we are still very far from being able to draw any positive conclusion about how CRISPR editing will allow to impact on telomere maintenance at one or another stage of the parasite life cell cycles. Therefore, the statement “how telomeres maintenance impacts the parasite's homeostasis” is a bit misleading for the readers, in that they might be disappointed when they get to the end of the review without having any hope concerning how to manipulate telomeres and telomerase to kill the parasite. In other words, there are still no telomeric pathways of Leishmaniathat could be targeted for efficient destruction of the parasite.

Minor comments:

* There is a good recent review on the use of vaccines and drugs for leishmaniasis. Maybe, the authors should discuss rapidly that point and cite it, Ghorbani M, Farhoudi R (2018). "Leishmaniasis in humans: drug or vaccine therapy?". Drug Design, Development and Therapy12: 25–40.

Reviewer 4 Report

This is a very interesting, timely, and informative review of telomeres and roles in Leishmania.  The authors have done an extensive literature review in general.  Some items that they should further consider are below:

  1. What is the reference for figure 1?
  2. italics for in vitro and in vivo through manuscript 
  3. There is only a limited amount of information about kinetoplast DNA (which is cancatenated)  about their telomers (do they have these??). It would be of interest to address this.
  4. the references in the bibliography are not done using consistent format (use of caps or not caps for article titles).
  5. Addition of a new figure (for section 4.1) to help visualize the TERT complex would be useful
  6. For lines 295 and 310:  since these are authors of the submitted manuscript, please indicate :  non published data rather than 'personal communication'
  7. some specific recommendations: 
  1.  line 24: remove s of telomeres 
  2. line 49: change 'being' to 'with'
  3. line 55: remove s and replace with n for awaken 
  4. line 57-58: remove 'parasite's drug' and add 'in these parasites' following targets
  5. for legend for figure 1 (line 59); use number in circle rather than brackets (which implies references)
  6. line 88 remove s from telomeres
  7. note that the references in Figure 2 legend are not in numerical sequence with the references in the body of the text. Same issue in Figure 3 legend.
  8. line 193:  change 'and' to 'but'
  9. line 144: remove 'to' and change be to being
  10. line 194: change 'is' to 'are' 
  11. line 217: change 'biding' to 'binding' 
  12. line 232:  change 'bind' to 'binding'
  13. line 247: remove 's'
  14. lines 251-260:  Base J or base J?  be consistent
  15. line 265:  change 'though' to 'thought'
  16. line 335: change assemble' to assembly'
  17. line 342:  the phrase '...would be a hard task..' is likely better as 'is not yet well understood' 
  18. line 345:  rephrase '...requires to be recruited to...' to '...must be recruited..'
  19. line 390: change 'are' to 'is'
